# Propolis, Bee Honey, and Their Components Protect against Coronavirus Disease 2019 (COVID-19): A Review of In Silico, In Vitro, and Clinical Studies

**DOI:** 10.3390/molecules26051232

**Published:** 2021-02-25

**Authors:** Amira Mohammed Ali, Hiroshi Kunugi

**Affiliations:** 1Department of Mental Disorder Research, National Institute of Neuroscience, National Center of Neurology and Psychiatry, Tokyo 187-0031, Japan; hkunugi@ncnp.go.jp; 2Department of Psychiatric Nursing and Mental Health, Faculty of Nursing, Alexandria University, Alexandria 21527, Egypt; 3Department of Psychiatry, Teikyo University School of Medicine, Tokyo 173-8605, Japan

**Keywords:** Coronaviruses, coronavirus disease 2019, COVID -19, severe acute respiratory syndrome, SARS-CoV-2, cytokine storm, propolis, bee honey, bee products, flavonoids, ACE-II, non-structural proteins, spike glycoprotein, main protease, in silico, in vitro, randomized clinical trials, molecular docking/biochemical modeling

## Abstract

Despite the virulence and high fatality of coronavirus disease 2019 (COVID-19), no specific antiviral treatment exists until the current moment. Natural agents with immune-promoting potentials such as bee products are being explored as possible treatments. Bee honey and propolis are rich in bioactive compounds that express strong antimicrobial, bactericidal, antiviral, anti-inflammatory, immunomodulatory, and antioxidant activities. This review examined the literature for the anti-COVID-19 effects of bee honey and propolis, with the aim of optimizing the use of these handy products as prophylactic or adjuvant treatments for people infected with severe acute respiratory syndrome-coronavirus-2 (SARS-CoV-2). Molecular simulations show that flavonoids in propolis and honey (e.g., rutin, naringin, caffeic acid phenyl ester, luteolin, and artepillin C) may inhibit viral spike fusion in host cells, viral-host interactions that trigger the cytokine storm, and viral replication. Similar to the potent antiviral drug remdesivir, rutin, propolis ethanolic extract, and propolis liposomes inhibited non-structural proteins of SARS-CoV-2 in vitro, and these compounds along with naringin inhibited SARS-CoV-2 infection in Vero E6 cells. Propolis extracts delivered by nanocarriers exhibit better antiviral effects against SARS-CoV-2 than ethanolic extracts. In line, hospitalized COVID-19 patients receiving green Brazilian propolis or a combination of honey and *Nigella sativa* exhibited earlier viral clearance, symptom recovery, discharge from the hospital as well as less mortality than counterparts receiving standard care alone. Thus, the use of bee products as an adjuvant treatment for COVID-19 may produce beneficial effects. Implications for treatment outcomes and issues to be considered in future studies are discussed.

## 1. Introduction

Coronavirus disease 2019 (COVID-19), announced by the World Health Organization as a global pandemic in March 2020, is a highly contagious viral infection caused by the newly discovered severe acute respiratory syndrome-coronavirus-2 (SARS-CoV-2) [1,2]. SARS-CoV-2 gets access to the human body through the respiratory tract causing severe acute pneumonia-associated respiratory syndrome (ARDS). It is rapidly transmitted among humans through droplet and direct contact [3,4,5,6]. To date, 23 February 2021, at least 111,419,939 confirmed cases of COVID-19 have been reported while global deaths reached 2,470,772 [7].

The cytokine storm is the main cause of fatalities of SARS-CoV-2 (around 15% of cases). It frequently occurs in old people, obese, diabetics, hypertensive, and those with cancer and cardiac dysfunction [8,9]. These individuals possess a baseline chronic state of subclinical inflammation associated with multiple aspects of immune deficiencies [10,11,12]. Evolving knowledge indicates that proper diet may promote immune resilience against COVID-19 [13,14]. Meanwhile, active searches are directed toward bioactive foods for the discovery of elements with anti-SARS-CoV-2-potential [10,15]. Propolis and bee honey, two well-known bee products, are used as dietary supplements and bioactive foods for nutrition and health promotion purposes [16,17,18,19,20]. This review summarizes in silico, in vitro, and clinical studies reporting on the anti-COVID-19 effects of propolis, natural honey, and their flavonoids.

## 2. SARS-CoV-2 and Associated Immune Response

SARS-CoV-2 is an enveloped positive-sense single-stranded RNA virus that belongs to beta coronaviruses, the Orthocoronavirinae family, Nidovirales order [2,5,9,21]. The lipid bilayer envelope of the virus comprises three structural proteins: the membrane (M) protein, the envelope (E) protein, and the spike (S) protein [5,9]. A helical capsid comprising nucleocapsid (N) protein contains the viral genome [5,21], which encodes structural and non-structural proteins (NSPs) [9,21]. For viral endocytosis, the cleavage of S protein by host serine proteases such as transmembrane protease serine 2 (TMPRSS2) is necessary, and it is followed by binding of the receptor-binding domain (RBD) in S1 subunit of S protein to angiotensin-converting enzyme-related carboxypeptidase (ACE-II) in host cells [2,22,23,24]. Papain-like protease (PLpro), RNA-dependent RNA polymerase (RdRp), and chymotrypsin-like protease (3CLpro), also known as main protease (Mpro), are NSPs that facilitate viral replication [9,21,25]. SARS-CoV-2 undergoes frequent evolutionary changes to facilitate its adaptation to novel human hosts [26,27,28]. Such changes may have implications for the development of vaccines and specific antiviral treatments [27,28,29].

Viral-ACE-II binding is followed by excessive signaling rewiring, which alters basic cellular processes (e.g., metabolism, antioxidant production, autophagy, etc.) and accelerates processes involved in cell cycle arrest [20,30,31,32,33]. PLpro, which is involved in viral replication, modulates signaling that alters immune defenses and contributes to the cytokine storm (e.g., nuclear factor kappa B and interferon 1) through its deubiquitinating (DUB) activity and removal of interferon-stimulated gene 15 from cellular proteins [34]. In addition, a few amino acid residues in S1 and S2 subunits of S protein, known as short linear motifs (SLiMs) LxxLxE, recruit protein phosphatase 2A (PP2A) by binding its B56 subunit [2]. PP2A modulates the activity of regulatory T cells (Tregs), and its dysregulation triggers Tregs dysfunction leading to uncontrollable release of inflammatory cytokines [35]. A wealth of studies reports remarkable alterations in different immune cells in COVID-19 patients, even during convalescence [11,12,35]. Immune cell dysfunction induced by SARS-CoV-2 is associated with a poor baseline immune functioning (immunological aging/inflammaging), and it is considered a key cause of multiple organ failure and fatality in severe COVID-19 [31,33,36,37].

## 3. Current Gaps in the Treatment of COVID-19 

Up to the current moment, there is no approved specific antiviral treatment for COVID-19. Therefore, several drugs (e.g., antiviral drugs, antibiotics, corticosteroids, and interferons) are repurposed for COVID-19 treatment without an evidence of efficacy in humans [5,6,38]. Anemia and diarrhea (common causes of malnutrition in COVID-19) are reported adverse effects of antiviral drugs (e.g., ribavirin and hydroxychloroquine) in previous coronavirus infections [38]. COVID-19 patients receiving antibiotics are at high risk for thrombogenic complications, liver damage, malnutrition, and hypoproteinemia [39,40,41]. Therefore, natural agents with multiple bioactivities are being explored for their potential to correct immune deficiency in COVID-19 and subsequently improve disease outcomes.

Molecular modeling and computational approaches—which predict the conformation of the ligand within the receptor and quantify the affinity of binding as a docking score (kcal/mol)—may allow a rapid and relatively cheap evaluation of the effectiveness of numerous compounds [5,42]. Given the documented antiviral, anti-inflammatory, antioxidant, and immunomodulatory activities of various bee products [19,31,33,43,44], we searched PubMed and Google Scholar for studies investigating their anti-COVID-19 effects using a combination of terms “propolis, bee honey, royal jelly, bee venom, bee pollen, phenolic compounds, flavonoids, coronavirus, SARS-CoV-2, COVID-19, in vitro, and randomized clinical trials.” Studies using active compounds in propolis and bee honey to target SARS-CoV-2 activities in silico, in vitro, and in humans were used for the synthesis of this narrative review. The few coming paragraphs describe the biostructure and bioactivities of propolis and bee honey.

## 4. Propolis Composition and Biological Activities

Propolis is a natural resinous substance produced by bee workers (*Apis mellifera*) from exudates collected from tree buds, which they mix with their saliva, pollen, and wax [20,32]. Propolis contains more than 300 identified compounds. It mainly comprises resins (50%), bee wax (30%), aromatic and essential oils (10%), bee pollen (5%), in addition to (5%) of multiple organic compounds such as polyphenols, flavonoids, amino acids, vitamins, and micronutrients [20,45,46,47]. Most active ingredients in propolis comprise polyphenolic phytochemicals that are widely distributed in vegetables and fruits: phenolic acids, flavonoids (flavanones, flavones, flavanols, etc.), stilbenes, and tannins [32,48,49]. Propolis contents of active compounds vary considerably according to its botanical origin and geographical location [32,47,50].

Caffeic acid phenethyl ester (CAPE) is a principal bioactive constituent in New Zealand and Egyptian propolis [51,52]. CAPE possesses a wide range of pharmacological properties e.g., anticancer, anti-inflammatory, antioxidant, etc., [20,32,52]. Artepillin C, a low-molecular weight single-ring phenol with 2 prenyl groups (3,5-diprenyl-4-hydroxycinnamic acid), uniquely exists in Brazilian green propolis (BGP) as a major bioactive ingredient. It contributes to the antioxidant, antimicrobial, and anticancer activities of BGP [2,52,53]. Both in vivo and in vitro studies show that artepillin C expresses strong anti-inflammatory effects during acute inflammation by inhibiting prostaglandin E (2) and nitric oxide through the modulation of major inflammatory signaling pathways such as nuclear factor kappa B [20,54]. Experimentally, a single oral dose of artepillin C decreased paw edema in rodents after 360 min of administration indicating its high bioavailability [20,54]. Characterization of the phenolic profile of Portuguese propolis through liquid chromatography with diode-array detection coupled to electrospray ionization tandem mass spectrometry revealed the presence of 76 polyphenols including typical poplar phenolic compounds (e.g., flavonoids and their methylated/esterified forms, phenylpropanoid acids, and their esters), as well as 14 glycosides of quercetin and kaempferol such as quercetin-3-*O*-rutinoside (rutin), quercetin-3-*O*-glucoside, kaempferol-3-*O*-rutinoside (nicotiflorin), isorhamnetin-3-*O*-rutinoside, and quercetin-3-*O*-rhamnoside)—some of these glycosides were never detected in propolis before [50]. Other bioactive compounds in propolis include *p*-coumaric acid, benzoic acid, galangin, pinocembrin, chrysin, and pinobanksin [51].

Thanks to its strong antimicrobial effects, bee workers use propolis for insulating hives, mummifying dead insects and animals that get into the hive, and for other disinfecting purposes [45,50]. For the same reason, propolis is widely used as a preservative of foods, beverages, and medicines [45,55]. Propolis is a key agent in apitherapy because of its rich flavonoid content as well as its antibacterial, antifungal, antioxidant, and anti-inflammatory properties [20,48,56,57]. Active ingredients in propolis express a strong immunomodulatory potential and potent inhibitory effects against numerous microbial enzymes (e.g., urease, xanthine oxidase, acetylcholinesterase, α-amylase, and α-glucosidase), which are involved in the pathology of several clinically important conditions [48]. The literature denotes a broad-spectrum antiviral activity of propolis, which involves inhibiting a variety of viruses such as herpes simplex virus, sindbis virus, parainfluenze-3 virus, human cytomegalovirus, dengue virus type-2, influenza virus A1, and rhinovirus [5]. RT-PCR denotes a superior antiviral activity of kaempferol, quercetin, chrysin, and luteolin to that of ribavirin against human rhinoviruses, which cause 50% of common colds attacks worldwide [58]. Kaempferol and *p*-coumaric acid remarkably reduced the RNA replication levels when administered early within 0–4 h after virus inoculation [58]. Propolis exerts a viricidal activity by destroying viral outer envelope (e.g., of human immunodeficiency virus); it also inhibits viral replication and transmission among cells [42]. Flavonoids of propolis are reported to exhibit potent ACE-inhibiting properties in vivo and in vitro [49,56].

Crude propolis is not suitable for human consumption because it is highly viscous and poorly dissolves in water. However, dissolving propolis in 60–80% ethanol results in an extract rich in most of its active polyphenols [20,48]. COOH moiety of ingredients of propolis such as caffeic acid and artepillin C impedes their penetration across the negatively charged plasma membrane resulting in less cellular permeability/bioavailability. However, Click Chemistry has been used recently to potently boost water solubility and cell permeability of these compounds by making their 1,2,3-triazolyl esters [59,60]. Aiming to develop an effective nanocarrier dosage to deliver both the hydrophilic and the lipophilic contents of propolis extract, Refaat et al. (2020) encapsulated Egyptian propolis extract within an optimized liposomal formulation of lipid molar concentration of 60 mM, cholesterol percentage of 20%, and drug loading of 5 mg/mL [51]. This formulation yielded nanosized particles (117 ± 11 nm) with entrapment efficiency% and released% of 70.1% and 81.8%, respectively [51]. Both pure BGP and its extracts should be protected against light and stored in a frozen state in order to retain their biological properties [61].

## 5. Bee Honey Composition and Biological Activities

Bee honey is a natural sweetener that comprises high amounts of reducing sugars, proteins, enzymes, amino acids, minerals, polyphenols, and vitamins [19,44,62]. In addition to its high nutritional value, honey has been long used for the treatment of several serious disorders because it enjoys a variety of pharmacological properties: anti-inflammatory, antioxidant, antidiabetic, anti-cancer, antilipidemic, antifungal, and bactericidal activities [37,63,64,65]. In addition, honey acts as a broad-spectrum antiviral agent, e.g., against varicella zoster virus and herpes simplex virus 1 (HSV-1). Indeed, in vitro and clinical trials show that honey can be used as an efficient alternative of acyclovir for the treatment of HSV-1 [66,67]. Moreover, folk medicine in many parts of the world considers natural honey as a first line of treatment for acute cough caused by upper respiratory tract infection, which is a key symptom in COVID-19 [68]. Methylglyoxal in honey interacts with microbial metabolites of different pathogens resulting in activation of mucosal-associated invariant T cells, which retain integrity of the mucosal barrier—nasal mucosa is the key access of SARS-CoV-2 to human body [69,70].Phenolic compounds in honey (flavonoids and phenols) contribute to most of its pharmacological properties [57,71]. Flavonoids in honey are classified into four groups: 1) flavonoles (e.g., kaempferol, fisetin, quercetin, galangin, and myricetin), 2) flavanones (e.g., hesperidin pinobanksin, naringin, and pinocembrin naringenin), 3) flavones (e.g., luteolin, genkwanin, apigenin, wogonin, tricetin, and acacetin), and 4) tannins (e.g., ellagic acid). It also comprises a large number of phenolic acids (gallic acid, p-hydroxybenzoic acid, caffeic acid, syringic acid, cinnamic acid, ferulic acid, vanillic acid, *p*-coumaric acid, chlorogenic acid, rosmarinic acid, and their derivatives), in addition to coumarins [71,72].

Honey promotes health by supporting the growth of healthy intestinal microflora and inhibiting the survival and activity of harmful endobacteria [19,37]. These effects are closely related to its low pH [62], high content of prebiotics such as oligosaccharides [73,74], and major species of beneficial lactic acid bacteria (LAB) such as *Bifidobacterium*, *Fructobacillus*, and *Lactobacillaceae* (e.g., *Lactobacillus kunkeei*—the most common species and the most potent in inhibiting pathogens highly resistant to antibiotics) [37,75]. Furthermore, 16S rRNA gene sequencing and MALDI-TOF along with UHPLC–ESI-MS/MS coupled to quadrupole orbitrap indicate that the shedding of cell membranes of dead LAB in honey (e.g., *Bacillus subtilis* and *Bacillus cereus*) represents the main source of its long-chain menaquinones content, subtypes of vitamin K_2_ [76], which demonstrate a range of health benefits (other than promoting coagulation) such as protecting against neurodegenerative disorders by contributing to the metabolism of myelin sulfatides in the brain [77]. Vitamin K_2_ potently binds SARS-CoV-2 proteins resulting in deficient binding to host cell receptors and inhibition of viral replication both *in silico* and *in vitro* [39].

## 6. Possible Anti-COVID-19 Effects of Flavonoids in Propolis and Bee Honey Reported by Molecular Docking Studies

Based on existing knowledge, efforts directed toward designing anti-COVID-19 drugs are focused on impeding viral entry into host cells, interrupting viral replication, and inhibiting viral-host protein interactions, with the aim of aborting the inflammatory responses induced by viral invasion [2,5,24,30]. As such, in silico studies have investigated the use of flavonoids in api-compounds as effective therapeutic candidates against COVID-19 by targeting S protein cleavage by host-cell proteases, e.g., TMPRSS2 [5], S protein binding to cell surface receptors such as ACE-II [48,51], inhibiting S protein [51,78], or S protein binding to the inflammatory B56 unit in PP2A [2] as well as by interfering with NSPs of SARS-CoV-2 in order to hamper viral replication [25,47,51]. Figure 1 provides an illustration of docking procedures. Further details are available in Appendix A.

Table 1 briefly summarizes the findings obtained from in silico studies, while Figure 2 presents the chemical structure of flavonoids noted in Table 1. This section sheds the light on the specific action of flavonoids derived from propolis and bee honey on targets of SARS-CoV-2 in silico.

### 6.1. Flavonoids in Propolis May Inhibit the Proteolytic Processing of S Protein by Host Proteases

While the cleavage of S protein by TMPRSS2 is essential for viral-host membrane fusion, viral proliferation may be blocked through TMPRSS2 inhibition by serine protease inhibitors [22,23], such as benzylsulfonyl-d-arginine-proline-4-amidinobenzylamide, peptide-conjugated phosphorodiamidate morpholino oligomers, and Camostat mesylate. The latter is commonly used to treat cancer, pancreatitis, and liver fibrosis [5,22]. In this respect, one study examined the interaction pattern of CAPE and two herbal extracts (withaferin and withanone) within the catalytic domain of TMPRSS2 compared with Camostat mesylate [5]. The binding affinity of CAPE to TMPRSS2 (−6.20 kcal/mol), involved hydrogen bond interactions with two main amino acid residues in the protease catalytic domain of TMPRSS2, Gly464 and Ser436 residues, in addition to several pi-pi interactions with other less influential residues. The inhibitory effect of CAPE was better than that of withaferin and withanone, and even Camostat mesylate (−5.90 kcal/mol), which all had a hydrogen bond interaction with only one residue, Gly464 [5].

### 6.2. Flavonoids in Propolis May Inhibit the Binding of SARS-CoV-2 to Host Cell Receptors

ACE-II is a type I integral membrane protein that functions as a metalloprotease enzyme. It contains 805 amino acids comprising one HEXXH-E zinc-binding consensus sequence in its active sites [48]. It is suggested that COVID-19 is highly contagious because SARS-CoV-2 easily enters human cells by binding human ACE-II more strongly than other coronaviruses [82,83,84]. The binding of S protein to ACE-II does not disrupt the ACE-II homodimer or ACE-II-RBD interfaces, while ACE-II homodimer conformations allow the binding of the spike trimer to more than one ACE-II homodimer [85]. Efforts toward designing agents that can express anti-COVID-19 properties focus on the use of molecules with efficient ACE-II-binding so that they compete with SARS-CoV-2 for ACE-II to block or delay viral entry into human cells [24,47,48].

In a study docking 43 flavonoids and three antiviral drugs to ACE-II, luteolin expressed a binding affinity (−10.1 kcal/mol) higher than that of hydroxychloroquine, comparable to Camostat mesylate and remdesivir (−7.7, −9.0, and −10.0 kcal/mol, respectively), see Figure 3b. The affinity of hydroxychloroquine for ACE-II was lower than other antiviral drugs and all flavonoids [86]. Another study compared the binding of 10 flavonoids copious in ethanolic extracts of propolis (Appendix A) to ACE-II with that of MLN-4760, a known blocker of ACE-II [48]. Rutin, CAPE, myricetin, quercetin, pinocembrin, and hesperetin (in order) expressed a binding affinity to ACE-II greater than that of the reference natural inhibitor molecule MLN-4760 (−7.28 kcal/mol and Ki of 4.65 μM), see Appendix A for details [48]. Rutin (Figure 4) expressed the highest binding potential (−8.97 kcal/mol and Ki = 0.261 μM) to zinc finger residues of ACE-II (Asn149, His345, Asp269, Glu375, Glu406, Thr371, Tyr127, and Asp368) via hydrogen bond interactions. It also demonstrated pi-cation interaction with Arg273, pi-pi T shape interaction with His374, alkyl interaction with Cys344, and pi-alkyl interaction with Tyr 127 residues [48].

### 6.3. Flavonoids in Propolis May Interrupt Viral-Host Protein Interactions That Induce an Inflammatory Response

The binding of SARS-CoV-2 to ACE-II is followed by a dramatic phosphorylation of viral-host proteins that promotes inflammation and cell cycle arrest [5,30]. Therefore, inhibiting S protein may cease viral entry into human cells [51,86]. Among 10 flavonoids docked into S protein, naringin (Figure 5) exhibited the highest binding affinity (−9.8 kcal/mol), even higher than dexamethasone (−7.9 kcal/mol), a standard drug repurposed for treating critically ill COVID-19 patients. Molecular dynamics simulation denoted conformational stability of naringin within the active site of S protein [78]. Quercetin was superior to other flavonoids (e.g., kaempferol and myricetin) and synthetic indole-chalcone blocking interaction sites on the viral spike Gly496, Asn501, Tyr505, and Tyr453 at an affinity of −7.8 kcal/mol [79]. Another study compared the binding of 10 bioactive compounds in Egyptian propolis (rutin, CAPE, quercetin, kaempferol, pinocembrin, pinobanksin, galangin, chrysin, *p*-coumaric acid, and benzoic acid) to S1 subunit with the binding of three repurposed antiviral drugs (Avigan, hydroxyquinone, and remdesivir) [51]. As shown in Figure 6, remdesivir exhibited the highest binding affinity to S1 followed by rutin and hydroxyquinone. Meanwhile, all other propolis-derived compounds (except benzoic acid) had a higher binding affinity than that of Avigan (Appendix A) [51].

Another direction for tackling COVID-19 is blocking viral-host protein interactions that induce acute inflammation—the key cause of tissue damage and disease fatalities. Altering viral binding to PP2A may possibly minimize the fatal effects of COVID-19 [2] because PP2A modulates immune cells and most cellular processes [11,12,35]. Artepillin C displays topological and structural similarities to leucine and glutamic acids, key residues in the side chains of two peptides located in S1 and S2 sub-unites of S protein, known as LxxIxE-like motif ^293^LDPLSE^298^ and ^1197^LIDLQE^1202^. These peptides interact with B56 regulatory subunit of PP2A (PP2A-B56) of host cells to activate inflammatory reactions [2]. Thanks to that similarity, leucine and glutamic acids in ^293^LDPLSE^298^ were superimposed with the two prenyl groups and acid group of artepillin C in silico [2]. Thus, artepillin C may act as a bioactive ligand inhibitor of S protein binding to PP2A [2]. Docking artepillin C and ^293^LDPLSE^298^ into PP2A-B56 revealed that both artepillin C and ^293^LDPLSE^298^ interacted with the same two pockets. Leu293 of ^293^LDPLSE^298^ interacted with the hydrophobic pocket of PP2A-B56 through hydrogen bonds, and its Glu298 residue formed ionic bonds with amino acid residues in the positively charged region of PP2A-B56. Artepillin C interacted with both pockets only through hydrogen bonds; however, at a binding affinity (−6.1 kcal/mol) greater than that of ^293^LDPLSE^298^ (−4.9 kcal/mol) indicating that artepillin C may compete with SARS-CoV-2 to bind PP2A-B56, which may possibly inhibit the acute inflammatory response associated with COVID-19 [2].

### 6.4. Flavonoids in Propolis and Honey May Interrupt SARS-CoV-2 Life Cycle

Given that alterations in NSPs such as 3CLpro/Mpro, PLpro, and RdRp interfere with viral replication and threaten viral survival [1,47,87], several studies have examined the interaction of flavonoids in propolis and honey with these key viral enzymes [25,42,47,51,86]. Preliminary results pinpoint an inhibitory activity of flavonoids against NSPs of SARS-CoV-2 [25,42,47,51]. The binding affinity of luteolin (−8.2 kcal/mol) to Mpro/3CLpro is reported to be greater than that of hydroxychloroquine, remdesivir, and Camostat mesylate (−5.4, −6.5, and −5.9 kcal/mol, respectively (Figure 3c) [86]. A similar trend was noticed when the four compounds were docked into PLpro: −7.1, −5.4, −6.5, and −5.9 kcal/mol, respectively (Figure 3d) [86]. Another study examined the binding of six active compounds in bee honey and propolis (3-phenyllactic acid, CAPE, lumichrome, galangin, chrysin, and caffeic acid) to the receptor active site of Mpro [47]. CAPE, chrysin, caffeic acid, and galangin interacted with Mpro through hydrogen bonds by binding their hydroxyl groups with amino acid residues Thr24, Thr26, Ser46, Gln189, and Hie164. The affinity of binding was high as depicted by glide scores of −6.386, −6.097, −4.387, −6.307 kcal/mol, respectively. In addition, all the four flavonoids formed pi-based electrostatic interactions with the receptor through Hie41 residue [47].

Rutin from Egyptian propolis had a stronger binding affinity to 3CLpro than hydroxyquinone, while different propolis extracts had a higher affinity than that of Avigan (Appendix A) [51]. Likewise, rutin formed stable intramolecular bonds with Mpro, particularly GLU 166, at a high affinity (−11.33 kcal/mol, Ki = 4.98 nM), comparable to that of theaflavin-3-3 (−12.41 kcal/mol, Ki = 794.96 pM) [88]. In a study docking 21 selected flavonoids to Mpro, rutin, comparable to remdesivir (−8.6 kcal/mol), demonstrated the highest binding (−8.7 kcal/mol) with the catalytic sites through hydrogen and electrostatic bonds. Of interest, remdesivir failed to make hydrogen bonds with the catalytic residues in the active sites of Mpro [81]. Pitsillou et al. [34] evaluated the blind docking of PLpro with 300 small molecules (phenolic compounds and fatty acids) and GRL-0617, a promising naphthalene-based noncovalent inhibitor. Rutin exhibited the highest inhibitory potential (−59.9 kcal/mol) relative to GRL-0617 (−36.3 kcal/mol), which was confirmed by time-dependent docking involving 100 μs molecular dynamics simulation trajectory of the PLpro as well as in vitro investigations (detailed below) [34]. Docking rutin in the naphthalene-inhibitor binding site of PLpro caused displacement of ubiquitin in a conformation that involves dysfunctional DUB activity [80].

Another study compared the affinity of binding of nicotiflorin, rutin, and their glucuronide and sulfate derivatives to 3CLpro and RdRp with that of X77 and theaflavin as positive controls. The affinity of binding of nicotiflorin (−11.2 kcal/mol) and rutin (−10.3 kcal/mol) to the main protease was close to that of X77 (−12.4 kcal/mol), while their affinity to RdRp (−11.3 and −11.7 kcal/mol, respectively) was better than that of theaflavin (−11.2 kcal/mol) [25]. The affinity of binding of metabolites of rutin (quercetin glucuronides, but not quercetin sulfates) to 3CLpro was better than that of rutin (ranging between −10.2 and −10.9 kcal/mol), and their binding to RdRp was also high (−10.2 to −10.5 kcal/mol). Among all nicotiflorin derivatives, kaempferol-7-glucuronide expressed the highest affinity of binding to 3CLpro (−10.9 kcal/mol), and the affinity of binding of all kaempferol glucuronides to RdRp was high (−10.0 to −10.2 kcal/mol). Kaempferol and quercetin were the least potent inhibitors of 3CLpro and RdRp among the derivatives of nicotiflorin and rutin. Most inhibitory effects of all compounds involved hydrogen bonds and pi-based interactions with protein residues of both 3CLpro and RdRp [25] (see Appendix A for further details). In a study docking several flavonoids and synthetic indole chalcones into the active sites of Mpro, quercetin was the second-best inhibitor candidate (−9.2 kcal/mol) following C23 indole-chalcone (−10.4 kcal/mol). The latter interacted with the protein at Glu288 and Asp289, while quercetin had an additional interaction at Glu290 [79].

Among 22 compounds found in Sulawesi propolis from Indonesia, broussoflavonol F and glyasperin A had a binding affinity of −7.8 kcal/mol to Mpro—greater than the affinity of potent inhibitors of beta coronaviruses known as α-ketoamide 13b and 14b (−8.2 and −7.2 kcal/mol, respectively) [42]. The interaction of both flavonoids with the protein involved hydrogen bond and hydrophobic interactions with amino acid residues in the catalytic sites (His41 and Cys145). A derivative of podophyllotoxin compounds known as sulabiroins A inhibited Mpro at an affinity of −7.6 kcal/mol through a hydrophobic interaction with the catalytic site His41 [42]. Altogether, flavonoids may inhibit NSPs of SARS-CoV-2 and interrupt viral replication [25,34,42,47,51,80,81,86].

## 7. Evidence from Experimental Studies Evaluating the Effect of Bee Products on SARS-CoV-2

A few studies have investigated the anti-SARS-CoV-2 effects of flavonoids in vitro. In one study, Vero E6 cells infected with SARS-CoV-2 at a multiplicity of infection (MOI) of 0.01 were treated with naringin (31.5, 62.5, and 250 μM). Naringin treatment inhibited SARS-CoV-2 infection and ameliorated the cytopathic effects of the virus compared with vehicle DMSO. The strongest effect was noticed with higher doses, with absence of cytotoxicity [89]. The human two-pore channel 2 (TPC2) plays a key role in the trafficking of the virions of coronaviruses to lysosomal compartments to allow priming of the S protein by lysosomal cysteine proteases (e.g., cathepsins B and L), which is followed by the release of the fusogenic peptide S2 [89,90,91]. Cells with knockdown of TPC2 by siRNA exhibited a strong inhibition of the replication of human coronavirus 229E by naringin signifying that naringin acts as an active lysosomotropic compound [89].

Two studies [34,80] compared PLpro inhibitory activity of rutin (3.1–200 μM) with that of GRL-0167 (100 μM) and hypericin (50 and 100 μM) in vitro using the proprietary papain-like protease (SARS-CoV-2 Assay Kit) for enzymatic inhibition assay. The percentage of inhibition of PLpro protease activity was 94% by GRL-0167, 97% and 87% by hypericin (50 and 100 μM, respectively), and 94% by rutin (100 μM) [34]. GRL-0167, hypericin, and rutin, in order, inhibited DUB activity in another study [80].

Refaat et al. [51] evaluated the inhibition of 3CLpro enzymatic activity (SARS-CoV-2 Assay Kit) by propolis extract, propolis liposomes, alcohol (vehicle), and remdesivir (500 µM). Both propolis extract and propolis liposomes inhibited 3CLpro; however, the inhibition induced by the optimized liposomal formulation was stronger (IC_50_ = 1.183 ± 0.06 µM) than that of propolis extract (IC_50_ = 2.452 ± 0.11 µM, *p* < 0.001). In Vero E6 cells infected with SARS-CoV-2 (MOI: 0.01), real-time PCR revealed a higher percentage of inhibition of SARS-CoV-2 replication by propolis liposomes relative to propolis extract (87.9% vs. 72.4%, *p* < 0.0001). The inhibitory effect of propolis liposomes against 3CLpro was comparable to that of remdesivir (91.2%) [51].

## 8. Evidence from Human Trials Evaluating the Effect of Bee Products on Patients with COVID-19

A retrospective Egyptian study described the use of oral TaibUVID—a combination of bee honey (15 mL), *Nigella sativa* (2 gm or 5 mL of *Nigella sativa* oil), and *Anthemis hyalina* (chamomile, 1 gm)—as an adjuvant or primary therapy in 13 and 7 COVID-19 patients, as well as a prophylactic agent in 20 individuals in contact with confirmed COVID-19 patients (doctors and family members) [92]. *Nigella sativa* and *Anthemis hyalina* are herbal plants that exhibit a plethora of health and immunity-promoting activities. Experimentally, these herbs, especially chamomile, efficiently lowered the survival of SARS-CoV in infected culture cells by altering intracellular Ca^2+^ concentration through the modulation of IL-8 expression and downregulation of a set of ion channel genes known as transient receptor potential proteins, resulting in potent alterations in viral cell functions [93]. The regular use of TaibUVID was associated with a rapid correction of lymphopenia, earlier symptom recovery in 70% of COVID-19 patients (within 1–4 days), and less occurrence of COVID-19 infection in contacts (70% vs. 30% of irregular users). The duration of treatment ranged between 2 and 20 days. Mild diarrhea, sweating, and hyperglycemia, which are evident in COVID-19 [40] were reported in three participants [92]. Because the participants used nebulizer containing a water extract of several herbs (*Nigella sativa*, *Anthemis hyaline,* and Costus) and prepared TaibUVID by themselves (different types of honey exist in Egypt, and the quality of herbs varies according to storage and other conditions), this retrospective design is unable to confirm the efficacy of honey under such circumstances [92].

We located a case study reporting symptom recovery and clearance of SARS-CoV-2 (indicated by negative PCR of a nasopharyngeal swab sample) following the consumption of three daily doses of a standardized non-alcoholic preparation of BGP (EPP-AF^®^, Apis Flora Industrial Comercial Ltda., Ribeirão Preto, Brazil) for 12 days in a 52-year-old woman positive for COVID-19 with mild symptoms [94]. However, a single case is less likely to provide a solid evidence of efficacy.

Some randomized controlled clinical trials (RCTs) are being conducted to assess the effectiveness of bee products in patients with a confirmed diagnosis of COVID-19. A Brazilian RCT reported that treating hospitalized COVID-19 patients with a single oral daily dose of EPP-AF^®^ was associated with significant reductions in the length of hospital stay (LOS) and renal injury. Propolis treatment was not associated with a decrease in the need for oxygen therapy [95]. Hospitalized Pakistani patients with moderate and severe COVID-19 received either an empty capsule (placebo) or a mixture of honey (1 gm/kg body weight/day) and *Nigella sativa* (80 mg/kg body weight/day) divided into 2–3 oral daily doses for 13 days [96]. Honey plus *Nigella sativa* (HNS) treatment was associated with a decreased time of symptom recovery in moderate and severe patients (4 vs. 7 days, hazard ratio (HR) = 6.11, 95% CI: 4.23–8.84, *p* < 0.0001) and (6 vs. 13 days, HR = 4.04, 95% CI: 2.46–6.64, *p* < 0.0001), respectively. Viral clearance was faster in moderate patients receiving HNS than in the placebo group (6 vs. 10 days, HR = 5.53; 95% CI: 3.76–8.14, *p* <0.0001). A 4-fold reduction in mortality was recorded in HNS-treated patients compared with patients receiving placebo (4% vs. 18.87%, odds ratio (OR) = 0.18, 95% CI: 0.02–0.92, *p* = 0.029). Severe patients in the HNS group achieved a mean oxygen saturation above 90% 6 days earlier than their placebo counterparts. The resumption of normal activity among moderate patients on day 6 was higher in HNS treatment than in placebo (63.6% vs. 10.9%, OR = 0.07, 95% CI: 0.03–0.13, *p* < 0.0001). The discharge of severe patients from the hospital at the end of treatment was higher in HNS treatment (50% vs. 2.8%, OR = 0.03; 95% CI: 0.01–0.09, *p* < 0.0001) [96]. Both RCTs reported absence of adverse effects following treatment with EPP-AF^®^ [95] and HNS [96]. As shown in Table 2, two clinical trials are underway. One study is being conducted in Egypt where non-severe COVID-19 patients would receive honey (1 gm/kg body weight/day) for 2 weeks [68]. The other trial involves treating COVID-19 patients with 300 mg of Iranian green propolis extract compared with a placebo capsule containing 300 mg microcrystalline cellulose. Patients will receive treatments three times a day for 2 weeks [97]. Results of these studies are awaited.

## 9. Discussion

Until now, there is no specific effective treatment for COVID-19. Efforts to control the pandemic of COVID-19 are directed toward inhibiting viral endocytosis and replication, preventing and repairing viral-induced tissue damage, and boosting host immunity. Molecular docking studies suggest that some flavonoids may play dual roles in targeting SARS-CoV-2, e.g., inhibiting TMPRSS2 [5], ACE-II [24], S protein, and NSPs [25,34,47,51,80,81] (Table 1). Therefore, these flavonoids may lower the chances of viral entry into host cells and decrease viral load and inflammatory reaction following infection. In vitro and human studies report congruent findings, albeit few studies are available.

Different docking programs exhibit variability in their ability to reproduce crystallographic binding orientations [98] denoting that reports on the binding affinity from different docking methods may not be directly compared. Nevertheless, there was some degree of consistency in the findings of different in silico studies suggesting that rutin, naringin, luteolin, CAPE, and quercetin may be potential inhibitors of SARS-CoV-2. Among several flavonoids, rutin expressed higher inhibition of S protein [51], ACE-II [48], and several NSPs of SARS-CoV-2 [25,34,51,80,81]. Various positive control inhibitors have been used in some docking studies such as natural and synthetic inhibitors of certain proteases [48,79] and antiviral drugs [51,86]. The latter are repurposed for COVID-19 treatment in clinical settings. Remdesivir expresses a stronger inhibitory effect on SARS-CoV-2 than other antiviral drugs such as Avigan [51], hydroxyquinone [51,86], and Camostat mesylate [86]. The effect of rutin was stronger than that of Avigan and hydroxyquinone—comparable to that of remdesivir [51,81]. Similarly, the inhibitory effect of luteolin against Mpro/3CLpro, PLpro, and ACE-II was comparable to Camostat mesylate and remdesivir [86]. CAPE inhibited TMPRSS2, ACE-II, and Mpro [5,47,48,51], while quercetin inhibited S protein and Mpro at an affinity higher than that of C23 indole-chalcone [79] and Avigan [51]. In a study docking quercetin and rutin to RdRp and Mpro, rutin had a considerably higher binding affinity than that of quercetin [25].

Flavonoids may also inhibit inflammatory signaling associated with SARS-CoV-2 through various mechanisms. Artepillin C exhibited a potent inhibitory effect on the binding of LxxIxE-like motifs, which exist in S1 and S2 subunits of S protein, to PP2A-B56 of host cells [2]. As such, artepillin C stands for a potent therapeutic agent that can regulate cellular functioning and grant protection against the cytokine storm induced by SARS-CoV-2 [2,35,99,100]. Likewise, inhibition of the DUB activity of PLpro by rutin both in silico and in vitro is likely to alter the inflammatory activity of this enzyme [80]. Naringin inhibited S protein in silico [78] and decreased viral load and related cytopathic effects in Vero E6 cells [89]. Meanwhile, a combination of honey and herbal plants is reported to correct lymphopenia in COVID-19 patients [92]. Indeed, flavonoids in honey and propolis can potently inhibit key inflammatory pathways, reduce oxidative stress, and protect against tissue damage [20,32,44,57]. The therapeutic effects of bee products are further enhanced when combined with extracts from herbal plants and exercise [37,57,70].

Identification of the most potent compounds is necessary for effective drug development; however, it is necessary to note that the pharmacological effects of supercritical extracts of BGP are superior to its single components (e.g., artepillin C and its precursor coumaric acid) indicating synergetic interaction of different compounds in propolis [52]. Experimental evidence shows that propolis liposomes express an inhibitory effect against SARS-CoV-2 similar to that of remdesivir [51]. In the same way, clinical trials report earlier viral clearance, faster symptom recovery, and reduced hospital stay in COVID-19 patients receiving a whole propolis extract or a mixture of natural honey and *Nigella sativa* [95,96]. These effects may be attributed to the activity of various flavonoids such as naringin [89] and luteolin [86] in honey as well as rutin and CAPE in propolis [48,51]. Therefore, future experimental studies may provide valuable information on the efficacy of the most bioactive compounds (e.g., rutin and naringin) as well as whole extracts of honey and propolis—alone and in combination with herbal extracts.

It is worth mentioning that liposomal formulations of propolis expressed a stronger inhibition of SARS-CoV-2 proteins/infection in vitro than the ordinary ethanolic extracts [51]. Therefore, using nanocarriers [51] and techniques known to improve water solubility of this compound (e.g., Click Chemistry) [59,60] may improve its delivery and enhance its functionality as a natural anti-COVID-19 treatment [51]. Removing allergens from propolis [101] and bee honey [102] may be necessary to avoid any possible adverse effects.

Despite the new insights provided by the findings reported in the current work, it is important to note that it included some preprints, which have not been subjected to any form of peer review [2,48,95,96]. In some of these preprints, vital information was missing in a few instances, e.g., on the nature of interactions of flavonoids with ACE-II receptor [48] and amino acid residues involved in the interaction of artepillin C with PP2A B56 [2]. In vitro and human studies are very few, and thus, more soundly designed studies are needed to confirm the effectiveness of these bee products in COVID-19 patients.

## 10. Conclusion

This review signifies a possibly valuable anti-COVID-19 potential of whole propolis liposomes and natural honey, as well as their flavonoids. In particular, rutin and naringin, along with other flavonoids, inhibited various SARS-CoV-2 proteins in silico. However, a few in vitro investigations are available, and they confirm the efficacy of rutin, naringin, and propolis liposomes against SARS-CoV-2. Among COVID-19 patients, propolis and combinations of bee honey with herbal plants were associated with improved viral clearance and symptom recovery along with earlier discharge from the hospital and decreased mortality. Future investigations should compare the effect of bee products alone or in combination with herbal plants as well as the effect of whole bee products and their key elements regarding their effects on oxidative stress, innate and adaptive immune system, inflammatory response associated with this virulent infection, and nutritional deficiencies in vulnerable groups.

## Figures and Tables

**Figure 1 molecules-26-01232-f001:**
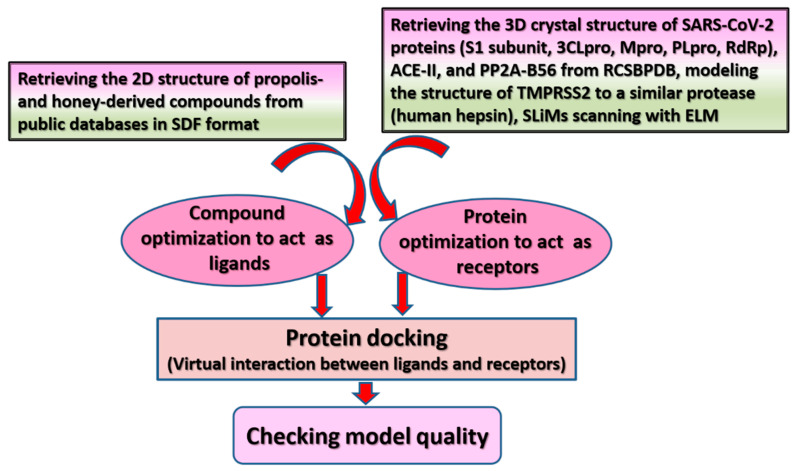
Schematic illustration of molecular modeling of interactions involving bioactive compounds in propolis and bee honey with SARS-CoV-2 proteins and host-cell receptor/proteases. 2D: two-dimensional; SARS-CoV-2: severe acute respiratory syndrome-coronavirus-2; 3CLpro: 3-chymotrypsin-like cysteine protease; RdRp: RNA-dependent RNA polymerase; PLpro: papain-like protease, ACE-II: angiotensin-converting enzyme-related carboxypeptidase; PP2A-B56: B56 regulatory unit of protein phosphatase 2 A; RCSBPDB: Research Collaboratory for Structural Bioinformatics Protein Data Bank; SLiMs: short linear motifs; ELM: eukaryotic linear motif resource. The two-dimensional (2D) or 3D structure of bee-derived compounds were obtained from ZINC database [25] (http://zinc15.docking.org/, accessed on 4 January 2021) or PubChem database (https://www.ebi.ac.uk/chembl/, accessed on 4 January 2021) [2,5,47,48]. Most studies retrieved the 3D crystal structures of ACE-II (PDB ID: R4L, the inhibitory bound state of the extracellular metallopeptidase domain of ACE-II with MLN-4760) [48], S1 subunit (PDB ID: 7BZ5) [51], PP2A-B56 (PDB ID: 5SWF-A) [2], and SARS-CoV-2 proteins such as S protein (PDB ID: 6m0j) [78] and (PDB ID: 7BZ5) [51], 3CLpro (PDB ID: 6LU7) [51], 3CLpro (PDB ID: 6Y2F, bound to α-ketoamide) [42], 3CLpro (PDB ID: 6W63, bound to ligand X77), Mpro (PDB ID: 5R7Y) [25], and RdRp (PDB ID:6M71) [25] from Research Collaboratory for Structural Bioinformatics Protein Data Bank (RCSBPDB) (http://www.rcsb.org, accessed on 4 January 2021) as PDB files. SLiMs (SARS-CoV-2 spike protein sequence) were scanned with the eukaryotic linear motif (ELM) resource (http://elm.eu.org/, accessed on 4 January 2021) [2]. Because the structure of TMPRSS2 was not available in RCSBPDB, one study modeled the structure of TMPRSS2 to a similar protease (human hepsin, obtained from the Swiss model repository (O15393)). The sequence of TMPRSS2 is 33.82% identical to hepsin; the catalytic domain in both proteins is well conserved with identical catalytic residues His296, Asp345, and Ser441, while the Q mean of the modeled structure was -1.62 [5]. Human cell enzymes (e.g., ACE-II and TMPRSS2) and SARS-CoV-2 proteins were geometrically optimized to act as receptors for virtual binding with bee-related compounds, which were also optimized to account for ligands in virtual binding models. The quality of docking was validated, and energy minimization was conducted [2].

**Figure 2 molecules-26-01232-f002:**
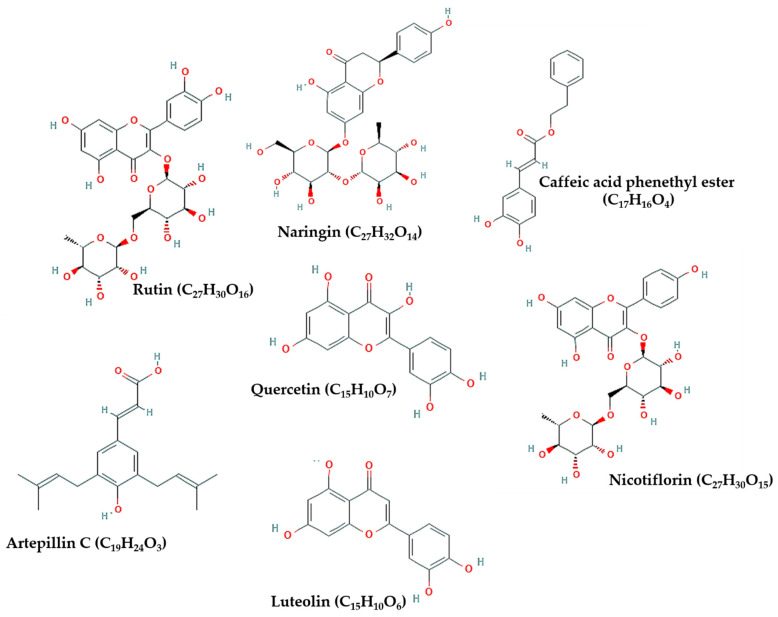
Chemical structure of flavonoids with the highest binding affinity to target proteins of severe acute respiratory syndrome-coronavirus-2 (SARS-CoV-2).

**Figure 3 molecules-26-01232-f003:**
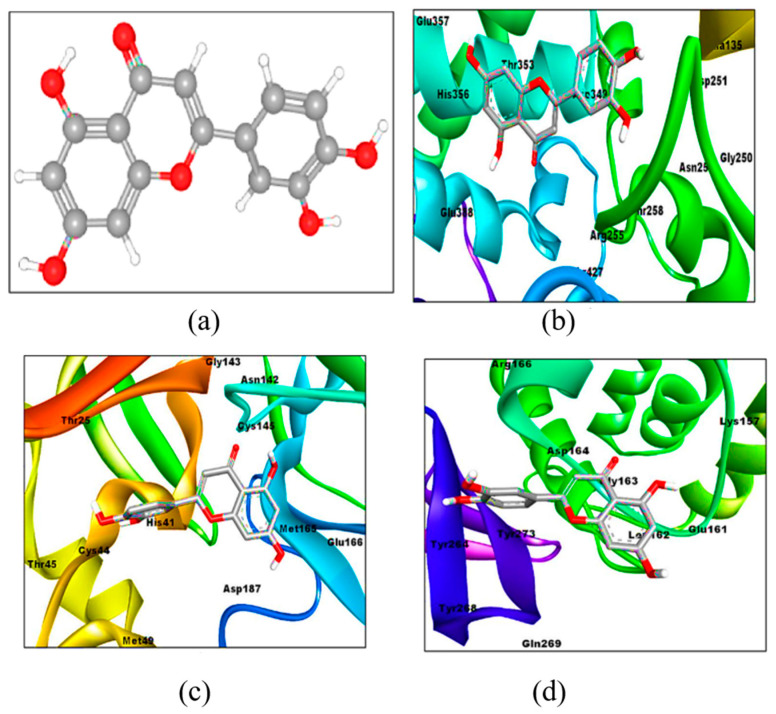
3D structure of luteolin (**a**) and binding poses involving its interaction within the active sites of ACE-II (**b**), Mpro/3CLpro (**c**), and PLpro (**d**). Modified with permission from Shawan et al. [86], Bulletin of the National Research Centre, Springer Open, 2021, http://creativecommons.org/licenses/by/4.0/.

**Figure 4 molecules-26-01232-f004:**
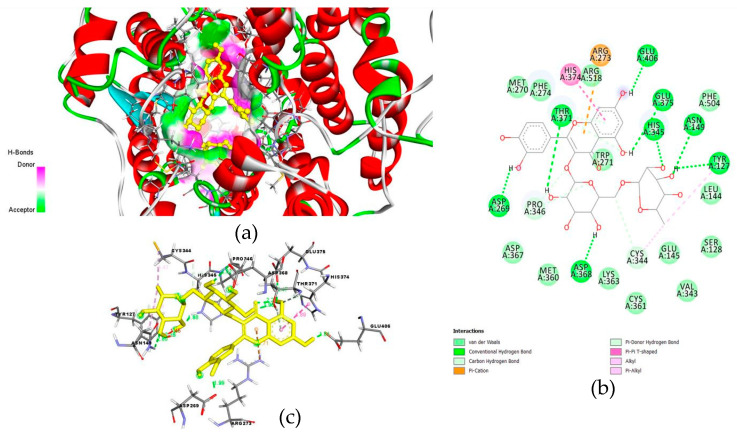
Binding pose (**a**) along with 2D (**b**) and 3D (**c**) analysis of molecular interaction of rutin within the active site of ACE-II. Reproduced with permission from Güler et al. [48], ScienceOpen Preprints, 2020, http://creativecommons.org/licenses/by/4.0/.

**Figure 5 molecules-26-01232-f005:**
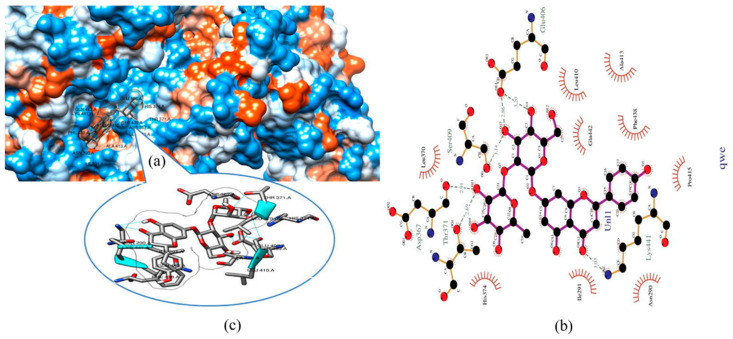
Binding pose (**a**) along with 2D (**b**) and 3D (**c**) analysis of molecular interaction of naringin within the active site of S protein. Reproduced with permission from Jain et al. [78], Saudi Journal of Biological Sciences, Elsevier, 2021, https://creativecommons.org/licenses/by-nc-nd/4.0/.

**Figure 6 molecules-26-01232-f006:**
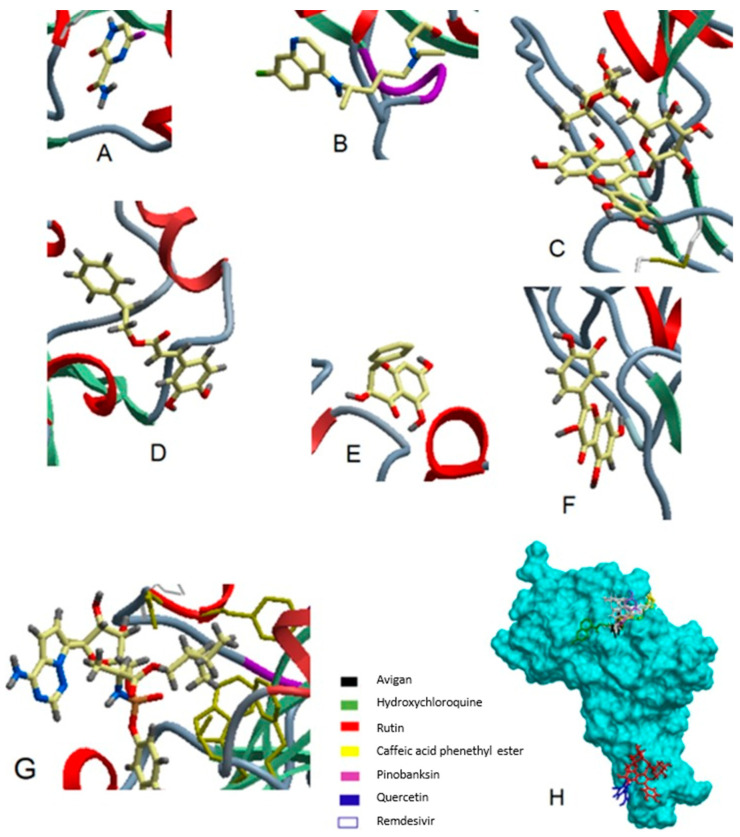
3D-plots for docking (**A**) Avigan, (**B**) hydroxychloroquine, (**C**) rutin, (**D**) caffeic acid phenethyl ester, (**E**) pinobanksin, (**F**) quercetin, and (**G**) remdesivir in the active site of S1 subunit of S protein, (PDB ID: 7BZ5). (**H**) 3D-plot comparing pose docking of Avigan and hydroxychloroquine to that of rutin, caffeic acid phenethyl ester, pinobanksin, and quercetin. Reproduced with permission from Refaat et al. [51], International Journal of Pharmaceutics, Elsevier B.V., 2020, https://creativecommons.org/licenses/by-nc-nd/4.0/.

**Table 1 molecules-26-01232-t001:** Flavonoids from propolis and bee honey exhibiting the highest binding to severe acute respiratory syndrome-coronavirus-2 (SARS-CoV-2) proteins and host cell receptor/proteases in silico along with their possible anti-Coronavirus Disease 2019 (COVID-19) effects.

Bee-Related Compounds	SARS-CoV-2 Proteins	Possible Anti-COVID-19 Effects	References
CAPE	TMPRSS2	Inhibiting S protein cleavage	[5]
Rutin, luteolin, and CAPE	ACE-II	Inhibiting viral binding to host cell receptor	[48,51,79]
Naringin, rutin, and quercetin	S protein	Inhibiting viral fusion in host cell membrane	[51,78]
Rutin, nicotiflorin, luteolin, and CAPE	3CLpro/Mpro, PLpro, and RdRp	Inhibiting viral replication and inflammatory reaction	[25,34,47,51,80,81]
Artepillin C	PP2A-B56	Inhibiting viral-host interactions that induce inflammation	[2]

TMPRSS2: transmembrane protease serine 2, CAPE: caffeic acid phenethyl ester, ACE-II: angiotensin-converting enzyme-related carboxypeptidase II, 3CLpro/Mpro: chymotrypsin-like protease/main protease, PLpro: papain-like protease, RdRp: RNA-dependent RNA polymerase, PP2A-B56: B56 regulatory unit of phosphatase 2A.

**Table 2 molecules-26-01232-t002:** Summary of studies treating COVID-19 patients with honey and propolis.

Bee Products	Sample and Design	Treatment	Study Outcomes	Results	References
Natural honey plus *Nigella sativa, Anthemis hyaline* (TaibUVID).	Confirmed COVID-19 patients (n = 20), contacts of COVID-19 patients (n = 20). Retrospective study.	TaibUVID orally and herbal solution inhalation.	Symptom recovery, blood count profile, and development of SARS-CoV-2 infection.	Improvement of the lymphocyte profile and earlier symptom recovery in regular users of TaibUVID. Lower incidence of SARS-CoV-2 infection in contacts.	[92]
BGP (EPP-AF).	Confirmed COVID-19 patient aged 52 years (n = 1). Case report.	EPP-AF^®^ 45 drops/3 times/day/2 weeks.	Early symptom recovery and viral clearance within 12 days of treatment.	Patient’s condition improved considerably. Negative nasopharyngeal swab (PCR).	[94]
BGP (EPP-AF).	Hospitalized adult COVID-19 patients (n = 82). An open-label, single center RCT.	EG1: propolis 400 mg (n = 40). EG1: propolis 800 mg (n = 42). EG1: standard care alone (n = 42).	LOS, dependence on oxygen therapy, development of acute kidney injury, ICU admission, use of vasoactive drugs.	Decreased LOS in EG1 and EG2 to 7 and 6 days compared with 12 days in CG. No effect of propolis on oxygen dependency. Decreased renal injury in EGs compared with CG (2 vs. 10 patients). No adverse effects of propolis were depicted.	[95]
Natural honey plus *Nigella sativa* (HNS).	Adults (n = 313) with moderate (n = 210) and severe (n = 103) COVID-19. A multicenter, placebo-controlled RCT.	EG: HNS (n = 107 moderate + 50 severe patients). CG: empty capsules placebo (n = 103 moderate + 53 severe patients).	Symptom recovery, viral clearance, a 30-day mortality, resumption of normal activity, oxygen saturation, and percentage of discharge from the hospital.	A 59% reduction in the time of symptom recovery in EG. Earlier virus clearance in EG. Decreased mortality by 4-folds in EG compared with CG. Higher resumption of normal activity on day 6 in moderate patients in EG than in CG. Achieving a mean oxygen saturation above 90% six days earlier in severe patients in EG. Higher hospital discharge of severe patients in EG. No adverse effects of HNS were depicted.	[96]
Iranian green propolis extract	Confirmed COVID-19 patients aged 18–75 years (n = 80). Double-blind, placebo-controlled, RCT	EG: propolis (n = 40). CG: placebo (n = 40).	COVID-19 severity and duration over 2 weeks.	NR	[97]
Natural honey	Non-severe COVID-19 patients aged 5–75 years (n = 1000). Single blind multicenter RCT.	EG: honey (n = 500). CG: standard care (n = 500).	Symptom recovery and viral clearance at day 14, lung recovery at day 30, mortality and viral clearance within 30 days.	NR	[68]

RCT: randomized clinical trial, EG: experimental group, CG: control group, BGP: Brazilian green propolis, LOS: length of hospital stay, HNS: honey plus nigella sativa, NR: not reported.

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
