# Peer review of "Propolis, Bee Honey, and Their Components Protect against Coronavirus Disease 2019 (COVID-19): A Review of In Silico, In Vitro, and Clinical Studies"

_molecules, 2021, doi:10.3390/molecules26051232_

Round 1

Reviewer 1 Report

The article attempts to review several recent publications that employ standard molecular docking techniques to identify potential interactions of some bee product ingredients with SARS-CoV-2 targets. Although the urgent need for efficacious anti-COVID agents is obvious, I am afraid that the article in its present form, unfortunately, fails to make a useful contribution in this area. Perhaps it could be published after a MAJOR REVISION if the authors would be able to address the following issues.

1) Although I would not go as far as to claim that purely computational predictions of biological activity are completely useless and should not be published, the review should clearly indicate if, and to what extent, any of those have been confirmed by experimental evidence and/or, at the very least, by more reliable computational approaches such as molecular dynamics. Apparently, for some compounds the molecular dynamics simulations actually indicate low activity. Taking into account the inherent uncertainty of molecular docking as well as the failures of almost all COVID-targeted drug repurposing attempts based on in silico or even in vitro predictions, a “review of unverified conjectures” is not only useless but might also be harmful for the scientific community and the Mankind in general because of increased informational noise in the time of this global crisis. This is exacerbated by the phrases like “capable of manipulating the main two mechanisms involved in the entry of SARS-CoV-2 into host cells” or “reported to effectively bind the main protease involved in the replication of SARS-CoV-2”, normally reserved for solid experimental data.

2) The article contains lots of excessive and wordy details that are only tangentially related to the topic, such as known data on SARS-CoV-2/COVID-19 genetics, molecular biology, and immunology (reproduced on 3 full pages), lists of target codes and amino acid interactions, or compound numbers in original articles. Most of these data should be significantly reduced, distilled to salient points, or moved to the supplementary materials.

3) On the other hand, the structures of the compounds under discussion, and preferably the figures of binding site interactions (if possible from the copyright perspective), should be present to enable meaningful reading.

4) Given that one of the most important complicating factors in COVID pathology is the progressive thrombosis and disseminated intravascular coagulation, the proposed beneficial effects of “vitamin k, a known coagulant approved by FDA”, require more careful discussion.

5) The units of measurement are incorrect or missing in some places, such as “drug loading of 5 mg/m” or “IC50 = 1.183 ± 0.06”.

6) The assertions such as “Evolving knowledge indicates that proper diet may promote immune resilience against COVID-19” or “honey is considered an efficient alternative for acyclovir for the treatment of HSV-1” should be supported by primary (experimental and/or clinical) literature.

7) English in the article is generally good, however, some misprints should be corrected.

Author Response

Manuscript ID: molecules-1084104.

Title: Protective effects of propolis, bee honey, and their components against coronavirus disease 2019 (Covid-19): a review of in silico, in vitro, and clinical studies

Response to Comments of Reviewer 1

We appreciate the Reviewer’s helpful comments and concern for clarity. The comments are addressed line-by-line as shown below. Replies come underneath in red.

First of all, we would like to express our gratitude to the reviewer for his/her deep and far-sighted comments, which really helped us refine this version of the manuscript.

  • Although I would not go as far as to claim that purely computational predictions of biological activity are completely useless and should not be published, the review should clearly indicate if, and to what extent, any of those have been confirmed by experimental evidence and/or, at the very least, by more reliable computational approaches such as molecular dynamics. Apparently, for some compounds the molecular dynamics simulations actually indicate low activity. Taking into account the inherent uncertainty of molecular docking as well as the failures of almost all COVID-targeted drug repurposing attempts based on in silicoor even in vitro predictions, a “review of unverified conjectures” is not only useless but might also be harmful for the scientific community and the Mankind in general because of increased informational noise in the time of this global crisis. This is exacerbated by the phrases like “capable of manipulating the main two mechanisms involved in the entry of SARS-CoV-2 into host cells” or “reported to effectively bind the main protease involved in the replication of SARS-CoV-2”, normally reserved for solid experimental data.

Authors’ response: We agree with the reviewer that the best evidence of efficacy would come from soundly designed clinical trials. Because of the novelty of COVID-19, RCTs of efficacy are generally lacking—including those testing the effect of bee products. Meanwhile, antiviral treatments of COVID-19 have not been denoted as successful yet. Given such uncertainty, in silico studies may act as a temporary guide for experimental/clinical trials, which produce valid results. Because we are keen not to provide dangerous information or be a source of informational noise, we modified our search strategy to include clinical and experimental studies that address the effect of bee products against COVID-19. Luckily, we found few studies. We have summarized the findings of human studies in Table 2 and provided a summary of in vitro studies on the effect of compounds in propolis on SARS-CoV-2 in Section 7. We have paid attention to our use of words expressing ideas in the manuscript so that they become neutral. We thank the reviewer for the insights provided by this comment.

  • The article contains lots of excessive and wordy details that are only tangentially related to the topic, such as known data on SARS-CoV-2/COVID-19 genetics, molecular biology, and immunology (reproduced on 3 full pages), lists of target codes and amino acid interactions, or compound numbers in original articles. Most of these data should be significantly reduced, distilled to salient points, or moved to the supplementary materials.

Authors’ response: According to this comment, we have removed most of the unnecessary text from different parts of the manuscript.

  • On the other hand, the structures of the compounds under discussion, and preferably the figures of binding site interactions (if possible from the copyright perspective), should be present to enable meaningful reading.

Authors’ response: Yes, we have obtained permissions to include some figures of binding site interactions involving flavonoids with the highest (as reported in consulted studies) binding affinity to SARS-CoV-2 proteins.

4) Given that one of the most important complicating factors in COVID pathology is the progressive thrombosis and disseminated intravascular coagulation, the proposed beneficial effects of “vitamin k, a known coagulant approved by FDA”, require more careful discussion.

Authors’ response: Thank you for raising such an important issue. However, reviewer 3 suggested reducing the discussion to its half size by focusing on the key points. So, many parts of the discussion were removed including the text relevant to vitamin k.

5) The units of measurement are incorrect or missing in some places, such as “drug loading of 5 mg/m” or “IC50 = 1.183 ± 0.06”.

Authors’ response: Yes, thank you very much. We have properly modified units of measurement lines [line 176 and 440].

6) The assertions such as “Evolving knowledge indicates that proper diet may promote immune resilience against COVID-19” or “honey is considered an efficient alternative for acyclovir for the treatment of HSV-1” should be supported by primary (experimental and/or clinical) literature.

Authors’ response: Yes, we have referred to primary (experimental and/or clinical) literature to support this information [line 57 and 188-190].

7) English in the article is generally good, however, some misprints should be corrected.

Authors’ response: We have carefully revised the whole manuscript to minimize language errors.

We hope that the comments were properly handled and that the revised version will be suitable for publication.

Best regards,

Reviewer 2 Report

This review describes the effects of compounds present in propolis and bee honey against SARS-CoV-2.

Section 2., Overview of SARS-CoV-2, such as sections 3,4,5, and 6  are very well described, understandably, with a review of recent literature data.

The problem has occurred with the review of molecular docking studies. There is a limited number of such studies since the COVID 19 is a novel disease, but six coronaviruses are known to infect human and causes mild to severe respiratory diseases. Plenty of inhibitors were developed against potential drugs against CoV-related diseases. Authors in this review report on the results of studies that are still not published, but there are published in form of unreviewed preprints.

These sections are based on preliminary reports that have not been peer-reviewed. They should not be reported in a scientific article as established information. Therefore, section 7 should be changed, deleted, or supplemented with scientific data from reviewed published studies. Also, the title of the paper should be changed, delete: “a review of in silico studies” is based on unreviewed reports.

Specific remarks:

  • page 2: in silico should write italic
  • page 2: Overview of SARS-CoV-2
  • page 7: References 55 and 32 are unreviewed study.
  • page 8: Section “Our literature….to “which is located in S1 subdomain of S protein [2].” is based on unreviewed reports and should be deleted from the manuscript.
  • page 8: Figure 1. is unclear. What procedure is presented by this scheme? Is it molecular docking? If it is, the scheme is incorrect. Protein modeling before generating 3D structures of compounds have no sense. Also, checking model quality before protein docking is incorrect. Figure 1 does not present a docking procedure correctly and should be deleted or changed.
  • page 9: The sentence: “Then, receptors were prepared, and model quality was checked before docking”. Which model and how is checked it quality? What does mean "prepared the receptor"? It is unclear.
  • page 9: Figure 2 is not an overview of scientific advances but possible positive effects of propolis and bee honey. Since those are only suppositions without scientific value, should be removed from the manuscript.
  • page 9: “because SARS-CoV-2 easily enters human cells by binding human ACE-II more strongly than other corona viruses “: (18) This is not the right reference. Please, provide original research if exists.
  • page 11: Table 1 is too extensive and should be shortened or transfer to the Supplementary files. Text in Table 1 is not the same font.
  • page 12: Do the compounds from reference (5) active compounds in propolis and bee honey?
  • page 18: P-Coumaric: P should be italic lowercase letter
  • page 20: withaferin and withanone: These two compounds are not phenols. They are secondary metabolites produced via the oxidation of steroids. Also, how is possible that these compounds could be constituents of honey and propolis?
  • page 20-21: Section: “Altering viral binding to…. of the acute inflammatory response associated with COVID-19 [2].” s based on preliminary reports that have not been peer-reviewed. They should not be reported in scientific articles as established information.
  • References should be written uniformly.
  • Reference 32: Journal name? Unfinished reference.
  • Reference 33: Journal should write as an abbreviation.
  • Reference 55: The name of the journal missing. This is published on ScienceOpen Preprints.

Author Response

Manuscript ID: molecules-1084104.

Title: Protective effects of propolis, bee honey, and their components against coronavirus disease 2019 (Covid-19): a review of in silico, in vitro, and clinical studies

Response to Comments of Reviewer 2

We appreciate the Reviewer’s helpful comments and concern for clarity as indicated by the provided comments. The comments are addressed line-by-line as shown below. Replies come underneath in red.

This review describes the effects of compounds present in propolis and bee honey against SARS-CoV-2.

Section 2., Overview of SARS-CoV-2, such as sections 3,4,5, and 6  are very well described, understandably, with a review of recent literature data.

The problem has occurred with the review of molecular docking studies. There is a limited number of such studies since the COVID 19 is a novel disease, but six coronaviruses are known to infect human and causes mild to severe respiratory diseases. Plenty of inhibitors were developed against potential drugs against CoV-related diseases. Authors in this review report on the results of studies that are still not published, but there are published in form of unreviewed preprints.

These sections are based on preliminary reports that have not been peer-reviewed. They should not be reported in a scientific article as established information. Therefore, section 7 should be changed, deleted, or supplemented with scientific data from reviewed published studies. Also, the title of the paper should be changed, delete: “a review of in silico studies” is based on unreviewed reports.

Authors’ response: Based on this comment section 7 has been removed.

We have expanded the search and included data from several other published studies. We have also reported on results from published in vitro and human studies. Accordingly, we have changed the title of this version to reflect on the nature of cited studies.

Specific remarks:

  • page 2: in silico should write italic

Authors’ response: Yes, in silico has been written italic

  • page 2: Overview of SARS-CoV-2

Authors’ response: We regret that this remark is unclear.

  • page 7: References 55 and 32 are unreviewed study.

Authors’ response: We appreciate the reviewer’s concern for accuracy. Because few studies are available, we included preprints. We referred to guidelines for authors submitting manuscripts to molecules, and found that they allows the inclusion of preprints as references: (https://www.mdpi.com/journal/molecules/instructions) "Unpublished data" intended for publication in a manuscript that is either planned, "in preparation" or "submitted" but not yet accepted, should be cited in the text and a reference should be added in the References section. "Personal Communication" should also be cited in the text and reference added in the References section. (see also the MDPI reference list and citations style guide).

  • page 8: Section “Our literature….to “which is located in S1 subdomain of S protein [2].” is based on unreviewed reports and should be deleted from the manuscript.

Authors’ response: Yes, this section has been removed to the Supplementary file.

  • page 8: Figure 1. is unclear. What procedure is presented by this scheme? Is it molecular docking? If it is, the scheme is incorrect. Protein modeling before generating 3D structures of compounds have no sense. Also, checking model quality before protein docking is incorrect. Figure 1 does not present a docking procedure correctly and should be deleted or changed.
  • page 9: The sentence: “Then, receptors were prepared, and model quality was checked before docking”. Which model and how is checked it quality? What does mean "prepared the receptor"? It is unclear.
  • Authors’ response to the two former comments: Thank you very much for insights provided by these two comments. The figure has been corrected accordingly.
  • page 9: Figure 2 is not an overview of scientific advances but possible positive effects of propolis and bee honey. Since those are only suppositions without scientific value, should be removed from the manuscript.

Authors’ response: The figure has been removed.

  • page 9: “because SARS-CoV-2 easily enters human cells by binding human ACE-II more strongly than other corona viruses “: (18) This is not the right reference. Please, provide original research if exists.

Authors’ response: Original references were cited as the reviewer recommended [line 274].

  • page 11: Table 1 is too extensive and should be shortened or transfer to the Supplementary files. Text in Table 1 is not the same font.

Authors’ response: That is right it is a huge table. It has been transferred to the Supplementary files, and a brief summary table is included in the manuscript.

  • page 12: Do the compounds from reference (5) active compounds in propolis and bee honey?

Authors’ response: No, they are not. They are referred to in the text as “two herbal extracts” [line 260].

  • page 18: P-Coumaric: P should be italic lowercase letter

Authors’ response: Yes, P has been changed to be an italic lowercase letter

  • page 20: withaferin and withanone: These two compounds are not phenols. They are secondary metabolites produced via the oxidation of steroids. Also, how is possible that these compounds could be constituents of honey and propolis?

Authors’ response: As noted in a former response, the manuscript refers to withaferin and withanone as “two herbal extracts (withaferin and withanone)” [line 260] not as constituents of honey and propolis. They are just mentioned for comparison same as Camostat, which were used in the same study.

  • page 20-21: Section: “Altering viral binding to…. of the acute inflammatory response associated with COVID-19 [2].” s based on preliminary reports that have not been peer-reviewed. They should not be reported in scientific articles as established information.

Authors’ response: Again, we appreciate the reviewer’s concern for accuracy; however, this is the only available report we could find on the interaction of bee-related compounds with PP2A in COVID-19 infection. Because the guidelines of Molecules ( https://www.mdpi.com/journal/molecules/instructions) allow the inclusion of these types of articles, we would keep it for the aforementioned reason. We have already warned the readers in the discussion and included the number of this reference on the list of unreviewed articles [Line 564].

  • References should be written uniformly.

Authors’ response: Yes, we have paid special attention to the uniformity of references.

  • Reference 32: Journal name? Unfinished reference.

Authors’ response: Yes, we have completed the reference. Thank you.

  • Reference 33: Journal should write as an abbreviation.

Authors’ response: We have abbreviated journal name.

  • Reference 55: The name of the journal missing. This is published on ScienceOpen Preprints.

Authors’ response: Yes, we have included ScienceOpen in this reference.

We hope that the comments were properly handled and that the revised version will be suitable for publication.

Best regards,

Reviewer 3 Report

The article by Ali and Kunugi assembled and combined the in silico studies conducted by many around the world. It is intriguing to see the number of studies that have been reported. Only one report has tested the compound on real coronavirus inhibition. 

Obviously the authors presented a comprehensive results of reported studies. However, at may places the authors have gone beyond the scope of the review article (as I assume that it is on CoV). The major weakness of the article is that it referred only in silico studies. The in silico studies are only good to a certain extent. Different methods provide different outcome and therefore, the binding affinity cannot be directly compared. 

In my opinion, it is worth publishing the review on natural compounds as their potential to inhibit the CoV. Having said that the article is needed to be shortened significantly.

Too much details of structural proteins have been presented. Knowing the fact the most of the said compounds are targeted to S protein. Hence,

  1. Section 2.3 should be reduced to by half or more.
  2. Same applies for Section 3. First 4 lines of 2nd paragraph on page 5 are useless.
  3. Similarly first two paragraphs of page 6 are needed to be significantly reduced.
  4. Table 1 has too many details. Why to include the compounds where the binding site is not given by the authors?
  5. Lat line on page 9 has incorrect statement. The amino acids do not inhibit the virus. It is the binding of compounds that does.
  6. Also, the active site of ACE2 is far away from the receptor binding domain. Please correct first sentence on page 10 or delete it.
  7. Reduce 'Discussion' section to half by just stating key points.

Author Response

Manuscript ID: molecules-1084104.

Title: Protective effects of propolis, bee honey, and their components against coronavirus disease 2019 (Covid-19): a review of in silico, in vitro, and clinical studies

Response to Comments of Reviewer 3

We appreciate the Reviewer’s helpful comments and concern for clarity. The comments are addressed line-by-line as shown below. Replies come underneath in red.

The article by Ali and Kunugi assembled and combined the in silico studies conducted by many around the world. It is intriguing to see the number of studies that have been reported. Only one report has tested the compound on real coronavirus inhibition. 

Obviously the authors presented a comprehensive results of reported studies. However, at may places the authors have gone beyond the scope of the review article (as I assume that it is on CoV). The major weakness of the article is that it referred only in silico studies. The in silico studies are only good to a certain extent. Different methods provide different outcome and therefore, the binding affinity cannot be directly compared. 

Authors’ response: We agree with the reviewer that in silico studies are not the best way to test the efficacy of therapeutic agents. We have quoted the statement given by the reviewer “Different methods provide different outcome and therefore, the binding affinity cannot be directly compared” in the discussion along with a suitable citation. Because experimental and human studies are likely to provide a stronger evidence, we have expanded our search and included Section 7 and Section 8 to report on findings from such studies, albeit we could obtain only few studies.

In my opinion, it is worth publishing the review on natural compounds as their potential to inhibit the CoV. Having said that the article is needed to be shortened significantly.

Authors’ response: Several available reviews describe the potential of bee products against COVID-19 in theory. We have expanded the current work hoping that it may complement previous works by synthesizing the findings from various studies that test the effects of honey and propolis as well as their flavonoids in silico, in vitro, and in human participants with COVID-19.

We have significantly shortened the article.

Too much details of structural proteins have been presented. Knowing the fact the most of the said compounds are targeted to S protein. Hence,

  1. Section 2.3 should be reduced to by half or more.
  2. Same applies for Section 3. First 4 lines of 2nd paragraph on page 5 are useless.

These lines have been deleted along with others.

  1. Similarly first two paragraphs of page 6 are needed to be significantly reduced.

Authors’ response to comments 1-3: According to these comments along with comments from other reviewer, we have removed most of the unnecessary text from different parts of the manuscript including the sections noted above.

  1. Table 1 has too many details. Why to include the compounds where the binding site is not given by the authors?

Authors’ response: Yes, we have moved the table to Supplementary material and replaced it by a briefer table.

  1. Lat line on page 9 has incorrect statement. The amino acids do not inhibit the virus. It is the binding of compounds that does.

Authors’ response: we agree with the reviewer the binding of flavonoids to amino acid residues distorts the function of viral proteins.

  1. Also, the active site of ACE2 is far away from the receptor binding domain. Please correct first sentence on page 10 or delete it.

Authors’ response: Yes, we have deleted the sentence as the reviewer suggested.

  1. Reduce 'Discussion' section to half by just stating key points.

Authors’ response: Yes, we have considerably reduced the 'Discussion'.

We hope that the comments were properly handled and that the revised version will be suitable for publication.

Best regards,

Round 2

Reviewer 1 Report

The authors have greatly improved the article, providing a balanced picture of the up-to-date in silico, in vitro, and clinical results. The article can be published after a MINOR revision addressing some presentation issues.

1) The structures of the key/representative compounds under discussion should be presented to enable meaningful reading.

2) The compound numbers used in the original articles (such as 13b, 14b, 15A, or 15C) are superfluous in a review, as is listing the entrapment efficiency and release percentage values to three decimal points (“70.112% and 81.801%”, line 170).

3) A number of misprints and stylistic inaccuracies should be corrected, such as “rich [in] bioactive compounds” (line 15), “t through” (line 42), “single positive-stranded RNA virus” (line 57, should be “positive-sense single-stranded RNA virus”), “alters processes involved in basic cellular processes” (line 71), “uncountable release” (line 80), “Until the current moment, there is no…” (line 87, should be “Up to the current moment, there is no…”), “water-solubility and cell-permeability” (line 163, hyphens are not needed). Subscripts or superscripts are missing in a few places (e.g., “IC50” or “Ca2+”). The reference list format is not completely consistent with the journal guidelines.

Author Response

Manuscript ID: molecules-1084104.

Title: Propolis, bee honey, and their components protect against coronavirus disease 2019 (Covid-19): a review of in silico, in vitro, and clinical studies.

Response to Comments of Reviewer 1

First of all, we would like to thank the reviewer for the precious time, hard effort, and sincere advice that he/she had kindly given to make this manuscript better. We have addressed the comments line-by-line as shown below. Replies come underneath in red.

First of all, we would like to express our gratitude to the reviewer for his/her deep and far-sighted comments, which really helped us refine this version of the manuscript.

  • The structures of the key/representative compounds under discussion should be presented to enable meaningful reading.

Authors’ response: Yes, we have included Figure 2 to present the molecular structure of flavonoids noted in Table 1.

  • The compound numbers used in the original articles (such as 13b, 14b, 15A, or 15C) are superfluous in a review, as is listing the entrapment efficiency and release percentage values to three decimal points (“70.112% and 81.801%”, line 170).

Authors’ response: We have limited the use of some of the noted compound numbers whenever possible. We have used only one decimal point.

3) A number of misprints and stylistic inaccuracies should be corrected, such as “rich [in] bioactive compounds” (line 15), “t through” (line 42), “single positive-stranded RNA virus” (line 57, should be “positive-sense single-stranded RNA virus”), “alters processes involved in basic cellular processes” (line 71), “uncountable release” (line 80), “Until the current moment, there is no…” (line 87, should be “Up to the current moment, there is no…”), “water-solubility and cell-permeability” (line 163, hyphens are not needed). Subscripts or superscripts are missing in a few places (e.g., “IC50” or “Ca2+”). The reference list format is not completely consistent with the journal guidelines.

Authors’ response: Thank you very much for the time the reviewer allocated for deep reading of the manuscript and detection of typos. We have corrected all the notes errors, and we used MDPI templet in Endnote to ensure compliance with the journal style of references. We have also checked all references manually to lower the chance of having mistakes. In few instances, the abbreviation of some journals e.g., Medico Legal Update was not available, so we left it as it is. Some papers do not have doi such as ref 63, 65, and 88. If the reviewer notices any problematic references in this version, I will be grateful if he/she indicates them.

We hope that we have satisfactorily modified the manuscript and that the revised version will be suitable for publication.

Best regards,

Reviewer 2 Report

The authors have corrected a manuscript according to the suggestions and now is ready for publishing.

Author Response

Manuscript ID: molecules-1084104.

Title: Propolis, bee honey, and their components protect against coronavirus disease 2019 (Covid-19): a review of in silico, in vitro, and clinical studies.

Response to Comments of Reviewer 2

We are much grateful for the time and help of the reviewer. Your comments really made a difference, thank you.

We have revised the language of the current version to have it with fewer errors.

Best regards,

Reviewer 3 Report

The authors have given serious consideration to address the concerns raised by the reviewers. I also agree with the limitations that authors have expressed. Additionally, my opinion is same as that of authors that, it is worth publishing the review on natural compounds as their potential to inhibit the CoV. 

Author Response

Manuscript ID: molecules-1084104.

Title: Propolis, bee honey, and their components protect against coronavirus disease 2019 (Covid-19): a review of in silico, in vitro, and clinical studies.

Response to Comments of Reviewer

Thank you very much for your considerate suggestions, time, and help. We are really indebted. We have attempted to revise the manuscript to ensure it is delivered in the correct language.

Best regards,
